# Identification of the Destruction Process in Quasi Brittle Concrete with Dispersed Fibers Based on Acoustic Emission and Sound Spectrum

**DOI:** 10.3390/ma12142266

**Published:** 2019-07-15

**Authors:** Dominik Logoń

**Affiliations:** Faculty of Civil Engineering, Wrocław University of Science and Technology, 50-377 Wrocław, Poland; dominik.logon@pwr.edu.pl

**Keywords:** acoustic emission AE, acoustic spectrum, quasi brittle cement composites, destruction process

## Abstract

The paper presents the identification of the destruction process in a quasi-brittle composite based on acoustic emission and the sound spectrum. The tests were conducted on a quasi-brittle composite. The sample was made from ordinary concrete with dispersed polypropylene fibers. The possibility of identifying the destruction process based on the acoustic emission and sound spectrum was confirmed and the ability to identify the destruction process was demonstrated. It was noted that in order to recognize the failure mechanisms accurately, it is necessary to first identify them separately. Three- and two-dimensional spectra were used to identify the destruction process. The three-dimensional spectrum provides additional information, enabling a better recognition of changes in the structure of the samples on the basis of the analysis of sound intensity, amplitudes, and frequencies. The paper shows the possibility of constructing quasi-brittle composites to limit the risk of catastrophic destruction processes and the possibility of identifying those processes with the use of acoustic emission at different stages of destruction.

## 1. Introduction

The application of acoustic emission (AE) measurements in determining the cracks, maximum load, and failure of reinforcement in cement composites has been widely presented in the literature.

The continuous AE evaluation in composites was earlier reported [1,2,3] and this technique has been applied to determine crack propagation in the fracture process in cement composites with and without reinforcement [4,5]. The acoustic emission (AE) events sum was also recorded for easier recognition of the first crack and crack propagation process [6,7,8]. 

It was also noticed that at the preliminary stage of degradation, the damage of the concrete elements was possible to detect with the application of the AE method [9,10]. The effectiveness of acoustic emission (AE) measurements in determining the critical stress of cement composites was tested [11], which enables the accurate definition of the elastic range corresponding to Hook’s law. Previously conducted tests have shown that AE is a good method for crack formation monitoring in mechanically loaded specimens [12,13,14,15,16,17,18] and has been successfully used to monitor structures [19,20]. Most of the papers have used AE to identify the destruction process of materials in structures [21,22,23,24,25,26,27,28] including crack orientation [29,30,31]. The AE method is still used and improved for the purpose of the identification of failure processes [32,33,34]. 

Previous works, however, have not focused on the correlation between AE and the individual failure processes of each of the different composite components based on the sound spectrum. These papers [10,11] showed that for the accurate recognition of composite failure processes, the AE (and the AE events sum) recording should be expanded to include the analysis of each sound separately and the analysis of the range of sounds corresponding to a given mechanical effect with the use of acoustic spectrum. The acoustic spectrum should be correlated with the load-deflection curve and with other acoustic effects, which enables the identification of the failure process (of the structure or the applied reinforcement) [10,11,18]. The quasi-brittle ESD cement composites (ESD—elastic range, strengthening, deflection control) are characterized by a higher load and absorbed energy in the elastic range when compared to the sample without reinforcement (E/E_0_) (Figure 2). Additionally, those composites are distinguished by a highly deflected structure damaged with macrocracks, multicracking effects, and the ability to carry additional stress in the strengthening area. Moreover, in the deflection control area, the samples’ ability to carry stress is higher than in the elastic range area. This paper focuses on determining the relation between the acoustic and mechanical ESD effects, in other words, reinforcement breaking, pull-out, macrocracks, and microcracking with the use of space spectrum. In [11], it was noted that in order to assess the destruction process, the analysis of a single signal and the AE events sum with the acoustic spectrum was required (each kind of the mechanical effect results in a different acoustic spectrum). In order to conduct a more in-depth analysis of the composite destruction process, what should be taken into account when interpreting the acoustic spectrum is not only the range of signals corresponding to a given mechanical effect (in a wide range of frequencies corresponding to the sound intensity), but also a single signal in a very small range of frequencies. 

It was confirmed that there is a possibility of correlation between AE and the failure process in ESD composites. That correlation enables a determination of the stage of damage in cement composites increasing the safety in the use of the composite and the decision of whether or not the damaged composite can be repaired. 

## 2. Materials and Methods 

The materials for the concrete (matrix—sample without reinforcement) consisted of: Portland cement CEM I 42.5R—368.7 kg/m^3^, silica fume 73.75 kg/m^3^, fly ash 73.75 kg/m^3^, sand and coarse aggregates 0/16 mm–1640 kg/m^3^, superplasticizer (SP), tap water 188.6 kg/m^3^, w/c = 0.51. 

The ESD concrete was reinforced with polypropylene fibers (curved/wave), minimum tensile strength 490 MPa, E = 3.5 GPa, equivalent diameter d = 0.8–1.2 mm, l = 54 mm. The reinforcement was randomly dispersed V_f_ = 1.5%.

Concrete was mixed in the concrete mixer and then used to mold samples. Beams (600 mm × 150 mm × 150 mm) were cast in slabs and then cured in water at 20 ± 2 °C. After 180 days of ageing, beams were prepared for the bending test (Figure 1). The samples were not notched.

Acoustic emission effects were recorded in order to monitor the progress of the fracture process in correlation with the load-deflection curve. The crosshead displacement was continuous and the rate was 1 mm/min. A seismic head HY919 (Spy Electronics Ltd.) was used to record the acoustic emission effects in the range from 0.2–20 kHz. The head was placed on the side in the central part of the loaded beams (Figure 1). The acoustic emission effects were presented as a 2D and 3D acoustic spectrum (amplitude of the frequency depending on sound intensity). The mechanical effects of the ESD composites were correlated with the recorded acoustic spectrum effects. The 2D sound spectrum was achieved with the use of the Audacity program (free digital audio editor) and the 3D spectrum using SpectraPLUS-SC (Pioneer Hill Software LLC, USA). 

Figure 2 presents the mechanical effects of the ESD (Eng. elastic range, strengthening, deflection control) cement composites with the corresponding acoustic effects and compiled acoustic spectra with various amplitudes corresponding to different mechanical effects (reinforcement breaking, pull-out, macrocracks, and microcracking). 

The ESD reinforcement effect is presented by characteristic points *f_x_*(*F_x_*-load, *ε_x_*-deflection, *W_x_*-work) and areas *A_X_* under the load-deflection curve.

## 3. Results 

Figure 3 presents the testing area for the four-point bending test with the AE acoustic emission measurements. Subsequent pictures show the characteristic stages of the ESD concrete failure process. Figure 3b indicates a crack occurring at the f_cr_ point, Figure 3c shows the multicracking (micro- and macrocracks), Figure 3d shows the progressing crack propagation, and Figure 3e shows the sample after the completed test. 

The load-deflection curve of the ESD composite and matrix (concrete without the dispersed reinforcement) is presented in Figure 4a. Above the curves, there are the results of the AE measurement with characteristic failure process events.

The ESD effects in the quasi-brittle composite were described with the use of the formula defining any points on the load-deflection curve f_x_ (load; deflection; absorbed energy). This formula enables the description and assessment of the ESD effects in the elastic range, strengthening, deflection control, and propagation areas. The matrix is characterized by f_max_ (Table 1).

For the ESD composite, the following results were achieved: f_cr_, f_max_, f_d_, (Table 1, Figure 4). Comparing the elastic range area of the ESD composite and the matrix, an x-time improvement was achieved for load, deflection, and absorbed energy A_E_/A_Ematrix_. A_S/E_ and A_D/E_ are the comparison of the strengthening A_S_ and deflection control A_D_ areas to the elastic range A_E_. The amount of absorbed energy in the strengthening area was considerably larger than in the deflection control area. The failure process propagation range following the deflection control area was not important in the ESD composites and was omitted.

Two-dimensional (2D) spectra of the matrix and ESD composite in the frequency range of 0–22 kHz are presented in Figure 4. Figure 4b shows the matrix spectrum for a crack f_cr_ = f_max_, additional spectra of the ESD sample for the first crack, f_cr_, and subsequent cracks, f_x1_, f_x2_, and f_max_. Figure 4c presents the ESD concrete spectra compared to the spectra of background noise, multicracking, and the fiber failure process. Three-dimensional (3D) spectra of the ESD composite are presented in Figure 5 and Figure 6.

The spectrum frequency range was limited to 200–6000 Hz as the greatest changes were observed within this frequency range in the sound spectra connected with the failure process. Figure 5a shows the background noise spectrum for the ESD composite recorded during the test, Figure 5b presents the first crack, f_cr_, and Figure 5c shows the multicracking. Figure 6a displays the macrocrack spectrum, Figure 6b shows the reinforcement destruction, and Figure 6c presents the fiber pull-out effect. 

## 4. Discussion

The existing provisions in the ASTM 1018 standard concerning the identification of characteristic points LOP, MOR, and ASTM indices I_5_, I_10_, I_15_ [35,36,37] have been extended by adding the possibility of describing any area or point f_x_ (load, deflection, energy). The introduced f_cr_, f_max_, and f_d_ points enable a precise description of the elastic range, strengthening, deflection control, and propagation areas as well as their comparison with one another with respect to the same sample or different samples.

The obtained results indicate that the ESD composite achieved the best effects in the strengthening area As. The improvement of properties in the elastic range AE was not good enough, which resulted from a low elasticity module of the fibers, causing a more significant deflection in this area. The deflection control range in the A_D_ area could be improved by increasing the fiber-matrix bond of the dispersed reinforcement. That effect may be achieved by increasing the strength of the composite or modifying the surface and geometry of the fibers.

2D and 3D spectra in the lowest frequency range did not record well (due to the head’s measurement range from 0.2 to 20 kHz) and were not taken into account in the interpretation of the results (2D spectrum 0–0.2 kHz). The 3D spectrum frequency range was limited to 200–6000 Hz. Within that frequency range, the greatest changes were observed in the sound spectra connected with the failure process.

Concrete without reinforcement (matrix) is characterized by a catastrophic destruction process. The appearing crack causes a destruction—breaking in halves—of the sample. The sound spectrum (within the range from −20 to −40 dB) corresponding to that process was positioned the highest when compared to other spectra characterizing various destruction processes, as shown in Figure 4a and is characterized by a small range of amplitudes. The background noise spectrum of the matrix has not been presented here, but was similar to the background noise spectrum of the ESD composite.

The ESD composite in the elastic range showed 2D and 3D spectra of background noise located low and within the range of the lowest amplitudes, as seen in Figure 4c and Figure 5a. The sound corresponding to f_cr_ was characterized by significant sound intensity, and the corresponding spectrum was located high, immediately below curve f_cr_ for the matrix, and the range of amplitudes was much larger (Figure 4a and Figure 5b). 

Subsequent cracks, f_x1_ and f_x2_, were situated at the level of background noise spectrum, but with the greatest amplitude range (Figure 4b).

The sound intensity of the multicracking was similar to background noise spectrum, but with a slightly greater range of amplitudes and significant amplitudes in a narrow frequency range (Figure 4c and Figure 5c).

Macrocracks showed the highest sound intensity. What is worth noting is the fact that the corresponding spectra were not characterized by the greatest amplitude range. The spectra were located the highest (Figure 4b).

The fiber pull-out process was characterized by a small range of amplitudes with a wide range of wavelengths (Figure 6c).

The sound spectrum corresponding to fiber failure was positioned low. The lower position of that spectrum when compared to that of the background noise may result from the manner of determining that spectrum. The background noise spectrum was determined with respect to a long period of time before the first crack and refers to a number of background noises in that period, whereas the fiber failure spectrum refers to a single signal. The sound spectrum for fiber failure was characterized by great amplitudes with a strong spike at 12–15 kHz. The average sound intensities of the fiber failure and the background noise were on a comparable level. The analysis of the background noise spectrum for a single sound in a short period of time resulted in a slight decrease in the sound intensity, but the amplitudes did not change significantly.

Frequency spike 12–15 kHz is an interesting, recurring correlation that may be used in the future for the identification of the failure process (Figure 4b,c). It is worth noting that that spike did not occur in the case of a catastrophic fracture f_cr_ (in a sample without reinforcement) in a short period of time. Frequency spike 12–15 kHz occurs in the case of defects generating acoustic effects that last for a longer period of time such as fiber failure, fiber pull out, and micro- and macrocracks that are blocked (stop propagating) or propagate slowly.

The conducted tests confirmed the possibility of identifying the failure process in traditional and ESD cement composites. The analysis of data showed that the 3D spectrum provided better general information for the identification of the failure process at each stage of the process, whereas the 2D spectrum enabled a more precise characterization of each of the sound spectra (sound intensity, amplitudes, frequency range) and their correlation with each of the failure processes. The simultaneous occurrence of failure processes makes their identification difficult. In order to differentiate them accurately, it is necessary to separately identify the sounds that do not overlap.

Summarizing the conducted tests, it can be stated that the analysis of 2D and 3D spectra is a good method of controlling the failure processes in the ESD cement composites. It increases the safety in the use of construction elements and enables correct decision-making in whether and how they should be repaired.

## 5. Conclusions

The research has allowed for the following conclusions to be formulated conclusions:(1)The 3D sound spectrum is a good tool for the observation and identification of failure processes in cement composites.(2)It has been noticed that the 2D spectrum enables a more precise identification and description of the sound spectra (sound intensity, amplitudes, frequency range) corresponding to different failure processes.(3)It has been suggested that for the analysis and identification of failure processes, both the 2D and 3D spectra should be used at the same time in a wide frequency range.(4)The development of ESD cement composites and the identification of failure processes with the use of AE and 2D and 3D spectra enables the control of failure processes (particularly useful in seismic areas or during natural disasters) or decisions of whether and how damaged cement composites should be repaired.

## Figures and Tables

**Figure 1 materials-12-02266-f001:**
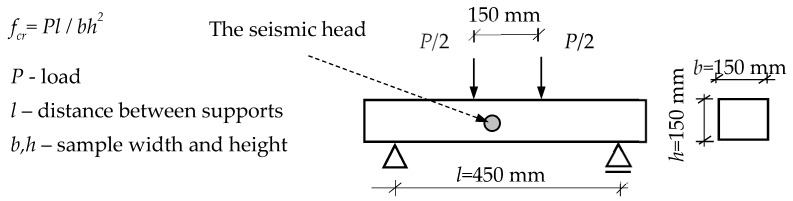
Four-point bending test.

**Figure 2 materials-12-02266-f002:**
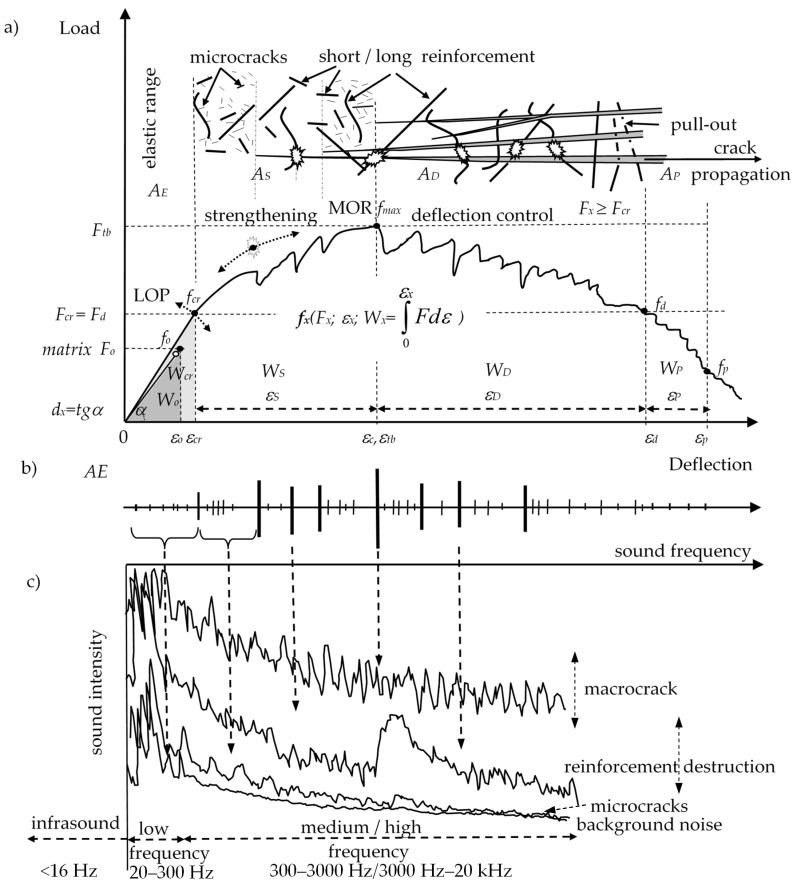
ESD composite: (**a**) load-deflection curve, (**b**) AE—acoustic emission effects, (**c**) 2D acoustic spectrum (frequency amplitude depending on sound intensity) [11].

**Figure 3 materials-12-02266-f003:**
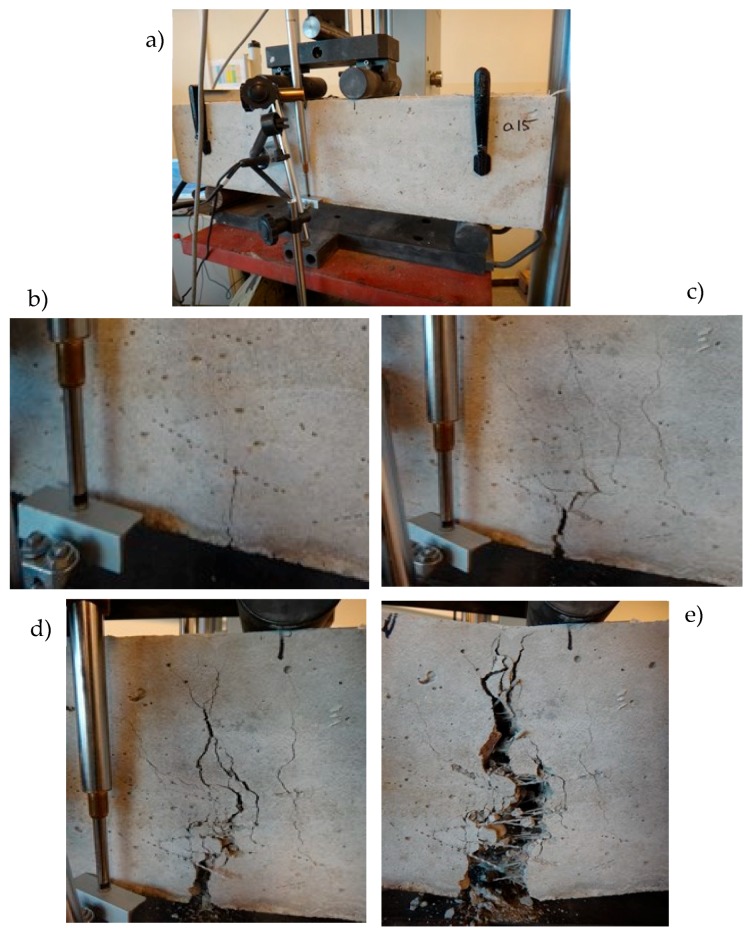
Four-point bending test: (**a**) sample before the test, (**b**) first crack at f_cr_ point, (**c**) multicracking (micro- and macrocracks), (**d**) destruction - propagation process, (**e**) view after the test.

**Figure 4 materials-12-02266-f004:**
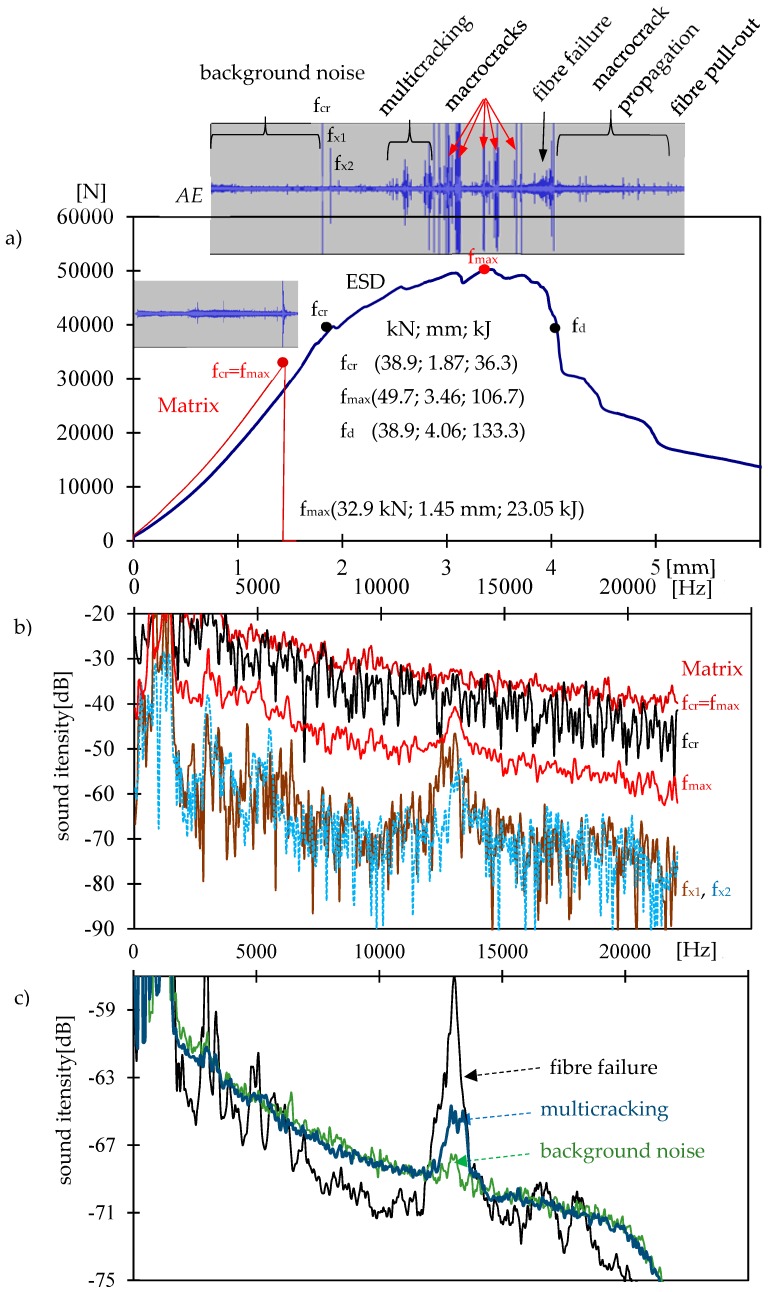
Matrix and ESD composite: (**a**) load-deflection curve, (**b**) 2D spectra of the matrix and ESD composite, (**c**) 2D spectrum of the ESD composite.

**Figure 5 materials-12-02266-f005:**
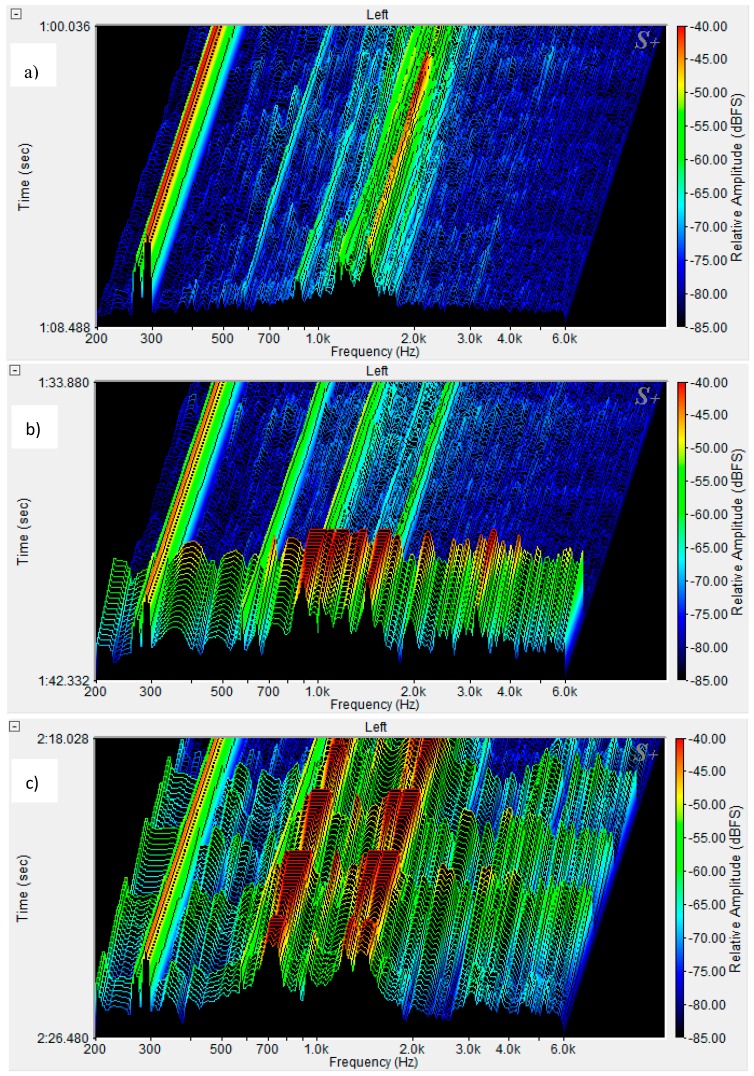
3D spectrum of ESD composite: (**a**) background noise; (**b**) first crack, f_cr_; and (**c**) multicracking.

**Figure 6 materials-12-02266-f006:**
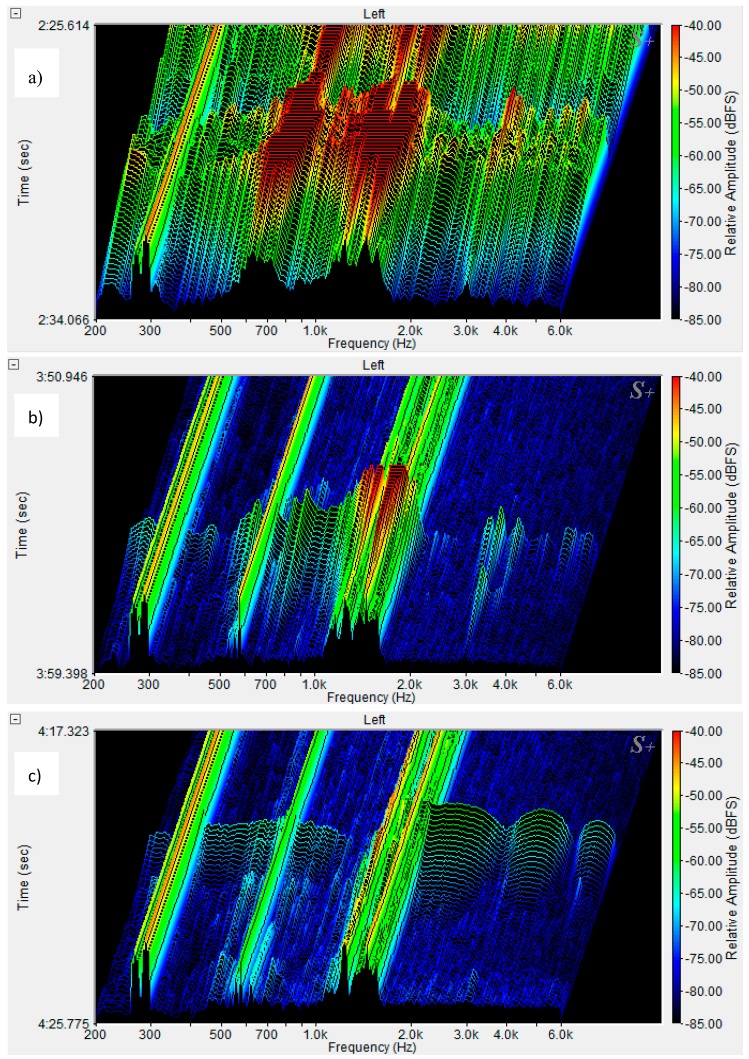
3D spectra of the ESD composite: (**a**) macrocrack, (**b**) fiber destruction, (**c**) fiber pull-out.

**Table 1 materials-12-02266-t001:** Mechanical properties of the matrix (concrete without reinforcement) and ESD composite.

Composite		Load F [N]	Deflection ε [mm]	Work W [kJ]	Ratio	Load	Deflection	Work
matrix	f_max_	32.9	1.42	23.0	-	-	-	-
ESD	f_cr_	38.9	1.87	36.3	A_E/Ematrix_	1.2	1.3	1.6
	f_tb_	49.7	3.46	106.7	A_S/E_	0.3	0.9	1.9
	f_d_	38.9	4.06	133.3	A_D/E_	-	0.4	0.7

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
