# Peer review of "Identification of the Destruction Process in Quasi Brittle Concrete with Dispersed Fibers Based on Acoustic Emission and Sound Spectrum"

_materials, 2019, doi:10.3390/ma12142266_

Reviewer 1 Report

The paper is presenting a method of using both 2D and 3D AE spectrums for improved damage detection and assessment in Concrete.  This idea is interesting, however, some points need to be clarified.

In the materials and Methods Section, the Authors state that a Seismic Head was used for the test but do not give the specification on it such as the frequency range.  This is needed to validate the frequency spectrum results.

Additionally, details on the data acquisition are not provided including sampling rate of the recorded waveforms.

I do not see any details on the loading method besides Figure 1 that gives the schematic. What is the loading rate applied?  This is critical as I expect the samples are rate dependent. 

Figure 2 shows a link between Load curve, AE waveforms, and frequency spectrum with failure modes.  How was this link established? Literature based or Experimental? Details are needed here to justify the conclusions that follow from this.

In the results section lines 140-142 appear to be left over from the template for the journal, please remove.

Figure 3 shows images of the failure evolution.  How were the images taken? an image every second or and image taken when AE activity changed?

Figure 4 shows an interesting and important plot, however I am confused about Figure 4c where a dominant frequency spike occurs between 10 and 15kHz.  Can you explain why this spike exists for all damage mechanisms examined? Typically there is a link between mechanism and frequency emitted. Particularly as a function of the type of material failed (cement, fiber) and size of the fracture.

Figure 5 and Figure 6 show the same thing with different frequency ranges? Can you be more clear about what each figure is showing?

Why not use a wavelet transform or a Short Time Fast Fourier Transform to examine the evolution of frequency content over time.  I feel like such a figure would be easier to understand and provide the same type of information as Figures 5 and 6.

Line 295 and 296 explains that Fig 5c shows multi cracking.  How was this connection established? Are you identifying damage mechanisms per part of the load curve and assuming all damage in that range is of one type? if so how did you verify a single mechanism was active and not multiple?

Why did you select to observe only to 6000 Hz in Figure 6? Explain more.

Line 357: the authors claim the head's measurement range is insufficient below 200 Hz, but have never given what the range actually is.

 Line 359 and 360: The authors claim that in the 200-6000Hz range the largest changes were observed in the sound spectrum from different damage processes. Can the authors explain in more detail the type of changes they are referring to.

Line 361 the authors write "Concrete without reinforcement (matrix)" this makes a lot of the document more clear in comparison, can the authors make this distinction earlier in the paper?

The Fiber destruction is stated to be lower than the background noise.  Can the authors give an explanation for this.  In most composite systems the fiber failure produces high frequency and high amplitude AE activity. Why is it different here?

The authors do not address the high attenuation that typically occurs in concrete systems. Can the authors use this to explain the frequency range used and justify the use of a single sensor to capture damage that occurs below the surface or on the back face that has to propagate a large distance to be recorded?  This would affect the frequencies observed and make identification more difficult. 

I agree that the possibility of identifying the destruction process is proven with this technique given the addition of the considerations mentioned previously.

Line 409 and 410: There is a statement stating this section is not mandatory, likely remaining from the journal paper template, please remove. 

Author Response

REPLY TO REVIEWER’S COMMENTS                                                                                      R#1

I am deeply grateful to the Reviewer for the effort put in the review of our paper.

I agree with most of the Reviewer’s comments and I have taken them into account in the paper’s revised version.

1) In the materials and Methods Section, the Authors state that a Seismic Head was used for the test but do not give the specification on it such as the frequency range.  This is needed to validate the frequency spectrum results.

The Seismic Head is designed for recording audible sounds in the range from 0.2-20 kHz. The information on the range has been added in the paper.

That range is confirmed by the analysis of 3D spectrums, which shows that sound spectrums in the range 0-0.2 kHz do not undergo significant changes during the recording of the destruction process, and above 22 kHz they are not recorded at all.

2) Additionally, details on the data acquisition are not provided including sampling rate of the recorded waveforms.

The sampling rate of the recorded waveforms was 44.1kHz. That information has been added in the paper. In the future, I intend to use 48kHz to verify the possibility of recording the acoustic effects more accurately.

3) I do not see any details on the loading method besides Figure 1 that gives the schematic. What is the loading rate applied?  This is critical as I expect the samples are rate dependent.

The crosshead displacement rate during the test was constant - 1mm/min. The information on the range has been added in the paper.

Along with the recording of crosshead displacement the acoustic emission was measured.

It is recommended to conduct the test much more slowly, as it will enable the observation/identification of the destruction process.

4) Figure 2 shows a link between Load curve, AE waveforms, and frequency spectrum with failure modes.  How was this link established? Literature based or Experimental? Details are needed here to justify the conclusions that follow from this.

That link was established on the basis of own tests of cement composites with various types of reinforcement. Some of the results that enable such a correlation were presented in an earlier paper [11], which is indicated in fig.2 by means of an appropriate literature reference.

5) In the results section lines 140-142 appear to be left over from the template for the journal, please remove.

This is an editorial error for which I am very sorry. This should not have taken place. The lines have now been deleted.

6) Figure 3 shows images of the failure evolution.  How were the images taken? an image every second or and image taken when AE activity changed?

The images were taken after the occurrence of characteristic noticeable microcracks, macrocracks or the noticeable process of fibre breaking or pull out. When taking the images, the time of taking an image was recorded, which allowed to link the images to the extent of bending (deflection) and recorded acoustic effects.

The loading test and AE measurement were started at the same time, which enabled the connection of noticeable destruction processes with the recorded acoustic effects.

It is recommended to continuously record the image of all surfaces of the sample (especially the bottom and the side surfaces).

7) Figure 4 shows an interesting and important plot, however I am confused about Figure 4c where a dominant frequency spike occurs between 10 and 15kHz.  Can you explain why this spike exists for all damage mechanisms examined? Typically there is a link between mechanism and frequency emitted. Particularly as a function of the type of material failed (cement, fiber) and size of the fracture.

Frequency spike 12-15 kHz is an interesting, recurring correlation that may be used in the future for the identification of the destruction process. 

 It is worth noting that that spike does not occur in the case of a catastrophic fracture fcr (in a sample without reinforcement). This fracture occurs in a short period of time. Frequency spike 12-15 kHz occurs in the case of defects generating acoustic effects that last for a longer period of time such as: fibre failure, fibre pull out, micro- and macrocracks that are blocked (stop propagating) or propagate slowly.

8) Figure 5 and Figure 6 show the same thing with different frequency ranges? Can you be more clear about what each figure is showing?

Figure 5 and figure 6 show different failure processes in the same frequency range 200-6000kHz

Figure 5. 3D spectrum of ESD composite: a) background noise, b) first crack fcr,                               c) multicracking.

Figure 6. 3D spectrums of ESD composite: a) macrocrack, b) fibre destruction, c) fibre pull-out.

9) Why not use a wavelet transform or a Short Time Fast Fourier Transform to examine the evolution of frequency content over time.  I feel like such a figure would be easier to understand and provide the same type of information as Figures 5 and 6.

A lot of researchers choose the data analysis method suggested by the reviewer. Such analyses were also conducted (but not included in the paper), as a test. In my opinion, 3D spectrum provides additional information facilitating the identification of various failure processes. Perhaps both methods may be used for the identification of the failure process?

10) Line 295 and 296 explains that Fig 5c shows multi cracking.  How was this connection established? Are you identifying damage mechanisms per part of the load curve and assuming all damage in that range is of one type? if so how did you verify a single mechanism was active and not multiple?. 

Multicracking is characterised by the occurrence of a number of microcracks which do not result in the decrease in the sample’s ability to carry stress. Multicracking is characterised by sound intensity similar to the background noise and the sound spectrums may overlap. 

In the case of the occurrence of a number of macrocracks, we observe each time a decrease on the load-deflection curve. The amplitude of sound intensity of macrocracks is much higher than in the case of the multicracking.

11) Why did you select to observe only to 6000 Hz in Figure 6? Explain more.

The greatest number of effects of 3D spectrum changes were observed in the range 200-6000Hz, Fig.5,6. If the range was increased to 20000Hz, those effects would be less visible (the effect of scale). The effects in the whole range to 20000 Hz are presented for 2D spectrums in fig.4.

12) Line 357: the authors claim the head's measurement range is insufficient below 200 Hz, but have never given what the range actually is.

The answer is provided in point 1.

 13) Line 359 and 360: The authors claim that in the 200-6000Hz range the largest changes were observed in the sound spectrum from different damage processes. Can the authors explain in more detail the type of changes they are referring to.

The refers to changes in sound intensity and amplitudes analysed in various frequency ranges.

14) Line 361 the authors write "Concrete without reinforcement (matrix)" this makes a lot of the document more clear in comparison, can the authors make this distinction earlier in the paper?

The explanation is provided in section “materials and methods”. The explanation has also now been added in the description of table 1.

15) The Fiber destruction is stated to be lower than the background noise.  Can the authors give an explanation for this.  In most composite systems the fiber failure produces high frequency and high amplitude AE activity. Why is it different here?

The difference results from the fact that Fig.4 presents background noise spectrum for a long period of time preceding the first crack, whereas the fibre failure spectrum refers to a single signal. The analysis of the background noise spectrum for a single sound in a short period of time results in a slight decrease in the sound intensity, but the amplitudes do not change significantly.

16) The authors do not address the high attenuation that typically occurs in concrete systems. Can the authors use this to explain the frequency range used and justify the use of a single sensor to capture damage that occurs below the surface or on the back face that has to propagate a large distance to be recorded?  This would affect the frequencies observed and make identification more difficult.

I agree that the possibility of identifying the destruction process is proven with this technique given the addition of the considerations mentioned previously.

The aim of the paper was to test the possibility of using the seismic head to record acoustic effects. 

Tests with the use of a larger number of heads with various parameters positioned in different places are certainly a good approach, but they require additional measurement equipment, which I intend to obtain in the future.

17) Line 409 and 410: There is a statement stating this section is not mandatory, likely remaining from the journal paper template, please remove.

The unnecessary lines have been removed.

The linguistic errors have been corrected. The paper has been checked by a sworn translator of the English language.

Author is convinced that many of the Reviewer’s suggestions will be helpful in further research and analyses which will form the basis for the next paper.

Once again I would like to thank the Reviewer most warmly for the perceptive and detailed comments, which greatly enhance the understanding of the paper and its value.

Reviewer 2 Report

Please see the attached file for comments.

Author Response

REPLY TO REVIEWER’S COMMENTS                                                                     R#2

I am deeply grateful to the Reviewer for the effort put in the review of our paper.

I agree with most of the Reviewer’s comments and I have taken them into account in the paper’s revised version.

Answers to the reviewer:

1. There is no need to write ‘AE’ in the title.

AE has been removed from the title.

2. All abbreviations i.e. 2D, 3D, ESD, AE, Vshould be avoided in the abstract. These short forms must be presented in the introduction first with full form. Later on, the short form should be used throughout the text.

2D, 3D, ESD, AE, Vf   have been removed from the abstract.

3. Please rewrite the abstract by mentioning the novelty you bring. This is not a summary. Avoid the sentence like “In summary it is stated….”

The abstract has been rewritten.

4. Line 45-46: “ These effects------reinforcement”. Correct this line.

The sentence has been removed.

5. Line 53-55: “It was noted--------“ is not clear. Please explain in detail and correct the text. There are many sentences like this in the manuscript. Authors are requested to use shorter sentences throughout the manuscript in order to avoid misleading meanings.

“It was noted--------“  -  It has been removed.

6. In Figure 1, ftb=Pl/bh3, define each term you used in the equation even if it is well known. Also define each term you used in Figure 2a, such as Fx, Wp etc. Moreover, the description of Figure 2 is very short and unclear. Explain in detail.

In Figure 1, explanations have been added.

Before Figure 2, the descriptions of point on the load-deflection curve  fx(Fx-load, ex-deflection, Wx-work) have been included in the text.

7. It is better to show the quantitative numbers as well, for low, medium and high frequencies in Figure 2c.

In Figure 2, the quantitative numbers for low, medium and high frequencies have been added

8. The discussion Section is unnecessarily long. Authors are requested to explain Figure 4, Figure 5 and Figure 6 with quantitative values in sequence to make it more understandable for the readers.

The quantitative values have not been given because the focus was on demonstrating the significant differences in spectrums corresponding to various destruction processes and not on describing those spectrums.

The remark is justified, but it would require a significant extension of the article.

9. There are no numerical simulations. What is the accuracy/uncertainty in measurements?

The quantitative values, numerical simulations and information on accuracy/uncertainty in measurements will be particularly required in the case of statistical papers with a large number of samples.

10. It seems that the author has written the manuscript in a very short time. Please check the Line 409-410.

I am very sorry, this is an editorial error that should not have taken place.

The linguistic errors have been corrected. The paper has been checked by a sworn translator of the English language.

Author is convinced that many of the Reviewer’s suggestions will be helpful in further research and analyses which will form the basis for the next paper.

Once again I would like to thank the Reviewer most warmly for the perceptive and detailed comments, which greatly enhance the understanding of the paper and its value.

Round  2

Reviewer 1 Report

Thank you for addressing my concerns.  I look forward to seeing your future studies

Reviewer 2 Report

All comments have been addressed in the modified version of the manuscript and it is suitable for the publication in the Journal of Materials.